# Structural and catalytic diversity of coronavirus proofreading exoribonuclease

Yu Li[1], Xiaocong Cao[2], Lauren M. Recker[1], Yang Yang [1] ✉ & Chang Liu [2] ✉

The coronavirus proofreading exoribonuclease (ExoN) is essential for genome fidelity and immune evasion of the viruses. Despite its critical roles in the viral life cycle, it is unclear how ExoNs across different coronaviruses diverge in their structures and catalytic properties, which may lead to differences in viral genome mutation rates and, consequently, viral fitness, immune evasion, and resistance to antiviral drugs. Here, we present comparative structural and biochemical analyses of ExoNs between two most representative human coronaviruses, Middle East Respiratory Syndrome Coronavirus (MERS-CoV) from the merbecovirus subgenus and SARS-CoV-2 from the sarbecovirus subgenus. Our results reveal a markedly lower catalytic activity of ExoN from MERS-CoV than that from SARS-CoV-2. The molecular basis of such a divergence across the two coronaviruses is unveiled by the cryo-EM structures of MERS-CoV ExoN in complex with RNA substrates bearing different 3′-end base pairs or mismatch, which represent the first set of ExoN structures from a coronavirus outside the sarbecovirus subgenus. Our findings also identify two highly conserved structural determinants that dictate efficient excision of different nucleotides at the 3′ terminus of RNA substrates by coronavirus ExoNs, a property that is pivotal for their roles in both viral RNA proofreading and immune evasion.

Coronaviruses will continue to have substantial societal, ethical, and economic impacts. To combat coronaviruses and associated diseases, it is imperative to thoroughly understand the mechanisms of coronavirus genome replication and transcription. Coronaviruses possess a single-stranded, positive-sense RNA [(+)RNA] genome[1,2], which is replicated and transcribed by a multi-subunit replication-transcription complex (RTC)[3]. The RTC comprises a series of viral encoded nonstructural proteins (nsp1–nsp16) harboring various RNA synthesis and processing enzymatic activities[3–5] and playing essential roles in the coronavirus life cycle. Among these viral nsps, the RNA polymerase nsp12 and its two accessory proteins nsp7 and nsp8 form a highly processive RNA-dependent RNA polymerase complex (RdRp)[6–9]. However, the viral RdRp is inherently error-prone and readily incorporates mismatches into newly synthesized RNA[10,11], which would lead to the accumulation of excessive deleterious mutations in the viral

genome and eventually viral extinction if left uncurbed. To mitigate the high error rate of RdRp-mediated RNA synthesis, coronaviruses have evolved a unique RNA proofreading mechanism mediated by the 3′-to-5′ exoribonuclease residing in the N-terminal domain of the bifunctional enzyme nsp14[12], which is structurally stabilized and functionally activated by its cofactor nsp10[13]. The C-terminal domain of nsp14 harbors a guanine N7 methyltransferase (N7-MTase) and catalyzes a critical step of reaction in the coronavirus RNA 5′ capping pathway[14,15]. Besides its RNA proofreading activity, the nsp10-nsp14 exoribonuclease complex (hereafter referred to as ExoN) also plays a critical role in degrading the viral double-stranded RNAs (dsRNAs)[16–18], which are intermediates of coronavirus genome replication and transcription[2], hence protecting them from being detected by the host innate immune sensors[19,20]. These essential roles of ExoN in the coronavirus life cycle highlight targeting ExoN as a novel and effective

[1]Roy J. Carver Department of Biochemistry, Biophysics and Molecular Biology, Iowa State University, Ames, IA, USA. [2]Department of Biophysics and Biophysical Chemistry, The Johns Hopkins University School of Medicine, Baltimore, MD, USA. ✉e-mail: yan9yang@iastate.edu; cliu207@jhmi.edu

strategy to intervene coronavirus replication[21,22]. The development of therapeutic strategies to effectively target or inhibit ExoN will greatly benefit from an in-depth understanding of its enzyme mechanism and substrate specificity.

Our previous work elucidated the detailed catalytic mechanism of SARS-CoV-2 ExoN mediated by the active site DEEDh motif and revealed how SARS-CoV-2 ExoN recognizes a dsRNA substrate containing a 3′-end U:C mismatch, providing mechanistic insight into its role in mismatch correction[23]. However, it remains to be seen how the ExoN recognizes and excises other types of nucleotides in RNA substrates, either in the context of mismatches or correct base pairs, which is important for the proper functioning of ExoN as both an RNA proofreader and a dsRNA-degrading immune suppressor[16,18].

ExoN is present in all coronaviruses known to date[12,24]. Although ExoNs in different coronaviruses are believed to share the general catalytic mechanism of RNA cleavage based on the conserved DEEDh catalytic motif, their sequences diverge to great extents in other regions. How such divergence in sequence affects the structures, substrate specificities, and catalytic activities of ExoNs from different coronaviruses is yet unknown, preventing the development of highly potent pan-coronavirus antivirals targeting ExoN.

Here, we determine the cryogenic-electron microscopy (cryo-EM) structures of ExoN from Middle East respiratory syndrome coronavirus (MERS-CoV), one of the three highly pathogenic human coronaviruses, in complex with various dsRNA substrates bearing either different 3′-end base pairs or mismatch at resolutions ranging from 2.5 to 2.9 Å (Supplementary Figs. 1–5; Supplementary Table 1). These high-resolution cryo-EM structures represent the first set of ExoN structures from a coronavirus outside the sarbecovirus subgenus and offer insights into how ExoN recognizes different 3′-end nucleotides of an RNA substrate. Together with an extensive comparative characterization of the catalytic activities and substrate specificities of ExoNs from MERS-CoV and SARS-CoV-2, we reveal a markedly lower catalytic activity of MERS-CoV ExoN than that of SARS-CoV-2 ExoN, which is explained by a critical difference in the active site conformations between the two ExoNs. In addition, our findings identify two highly conserved structural determinants that dictate the broad substrate specificity of coronavirus ExoNs by allowing the proper recognition of different nucleotides in RNA substrates by ExoNs, a property that is pivotal for their roles in both RNA proofreading and immune evasion of coronaviruses.

## Results

### Structure and catalytic activity of MERS-CoV ExoN
Among all coronaviruses, three-dimensional structures of ExoN are available for only those from SARS-CoV and SARS-CoV-2[15,23,25–27], both of which belong to the sarbecovirus subgenus and whose ExoNs share a 95% sequence identity. Between the two, RNA substrate recognition has been visualized structurally solely for the SARS-CoV-2 ExoN[23]. To provide the insight into the structural and catalytic diversity of coronavirus ExoNs, we systematically studied the ExoN from MERS-CoV, a coronavirus in the merbecovirus subgenus and of major clinical significance. Notably, MERS-CoV ExoN exhibits considerable sequence divergence from SARS-CoV-2 ExoN (37% sequence difference between the two ExoNs). Despite their sequence divergence, MERS-CoV ExoN complex has the same domain organization and a completely conserved DEEDh catalytic motif as SARS-CoV-2 ExoN does (Fig. 1a). To examine the catalytic activity of MERS-CoV ExoN, we assessed its digestion of a hairpin RNA (hereafter referred to as T20P15) that comprises a 20-nucleotide (nt) template strand (T-RNA) region and a 15-nt product strand (P-RNA) region (Fig. 1b). The T20P15 RNA mimics an intermediate product of RNA replication from a 3′-end fragment of the MERS-CoV genome but contains a 3′-end U:C mismatch. Wild-type (WT) MERS-CoV ExoN actively digests the T20P15 RNA, whereas the ExoN bearing a

catalytically inactive mutation E191A (hereafter referred to as ExoN E191A) shows no RNA cleavage activity on the substrate (Fig. 1c). To reveal the molecular details of RNA substrate recognition by MERS-CoV ExoN, we reconstituted and purified the complex of ExoN E191A mutant with the T20P15 RNA, followed by single-particle cryo-EM analysis (Supplementary Fig. 1a–c). The final cryo-EM map for the MERS-CoV ExoN•T20P15 complex was refined to 2.7 Å (Fig. 1d and Supplementary Fig. 1c). Similar to that observed in the SARS-CoV-2 ExoN•RNA complex structure[23], MERS-CoV ExoN captures the 3′-end +1C$_P$, breaking its mismatched pairing with +1U$_T$ (nucleotide numbering shown in Fig. 1b). Subscripted $_P$ and $_T$ denotes P-RNA and T-RNA strands, respectively), which is largely disordered in the MERS-CoV ExoN•T20P15 RNA complex structure (Fig. 1e). In addition, the shallow dsRNA-binding pocket of MERS-CoV ExoN interactions with the backbone of the RNA only at −1 and −2 positions through multiple hydrogen bonds and salt bridges (Fig. 1e), while leaving majority part of the RNA duplex in the solvent-accessible space (Fig. 1d). In MERS-CoV ExoN active site, the scissile phosphate between +1C$_P$ and −1C$_P$ of the RNA substrate is coordinated by a Mg$^{2+}$ ion, which is in turn chelated by three catalytic carboxylate residues of ExoN (D90, E92, and D273 of nsp14) (Fig. 1f). A second divalent cation essential for the catalysis of RNA digestion is missing from the active site of ExoN in our structure (Fig. 1f), most likely due to the absence of E191 side chain carboxylate. Such a defect in the ExoN active site structure was also observed in SARS-CoV-2 ExoN E191A•RNA complexes[23] and explains the lack of catalytic activity of the MERS-CoV ExoN E191A mutant (Fig. 1c).

Our cryo-EM data also yielded a dimeric form of the MERS-CoV ExoN•T20P15 complex, which exhibits an almost identical overall architecture of ExoN and pattern of RNA recognition compared with the monomeric form of the complex (Supplementary Fig. 1c–e). The dimerization is mainly mediated by two pseudo-symmetric interfaces that bury 641 Å2 and 416 Å2 of solvent-accessible surface, respectively (Supplementary Fig. 1d).

### Structural and catalytic divergence of MERS-CoV and SARS-CoV-2 ExoN
Although the sequences of ExoNs from MERS-CoV and SARS-CoV-2 exhibit considerable differences, the overall structure of MERS-CoV ExoN•RNA complex is highly similar to the SARS-CoV-2 counterpart (Fig. 2a). The superimposition of the two ExoN•RNA complexes, however, also revealed a key difference in their active site conformations (Fig. 2b). The nsp14 α4-α5 loop harboring H268, the fifth catalytic residues of the conserved ExoN DEEDh motif, adopts two distinct conformations in MERS-CoV and SARS-CoV-2 ExoNs, resulting in a 3.3 Å shift of H268 away from the scissile phosphate of substrate RNA in the MERS-CoV ExoN•RNA complex compared with that in the SARS-CoV-2 ExoN•RNA complex (Fig. 2b). Whereas RNA binding to SARS-CoV-2 ExoN, either the WT or E191A mutant form, induces a substantial structural rearrangement of its H268-harboring α4-α5 loop, facilitating the full assembly of the ExoN active site[23], the conformation of the α4-α5 loop in the MERS-CoV ExoN•RNA complex resembles that in the RNA-free apo SARS-CoV ExoN[15,25] (Fig. 2c), which represents an inactivated state prior to the binding of RNA substrate[23]. In addition, we modeled the position of the catalytic water molecule in the MERS-CoV ExoN•RNA complex structure based on its superimposition with the SARS-CoV-2 WT ExoN•RNA complex structure[23] (Fig. 2d). Our structural modeling suggests that H268 of MERS-CoV ExoN would be excessively distant from the catalytic water (>4 Å) (Fig. 2d), hampering an efficient deprotonation of the catalytic water molecule for subsequent nucleophilic attack of the scissile phosphate[23].

This improper positioning of MERS-CoV nsp14 H268 suggests a weakened catalytic activity of MERS-CoV ExoN compared with that of SARS-CoV-2 ExoN. To assess the catalytic difference between these

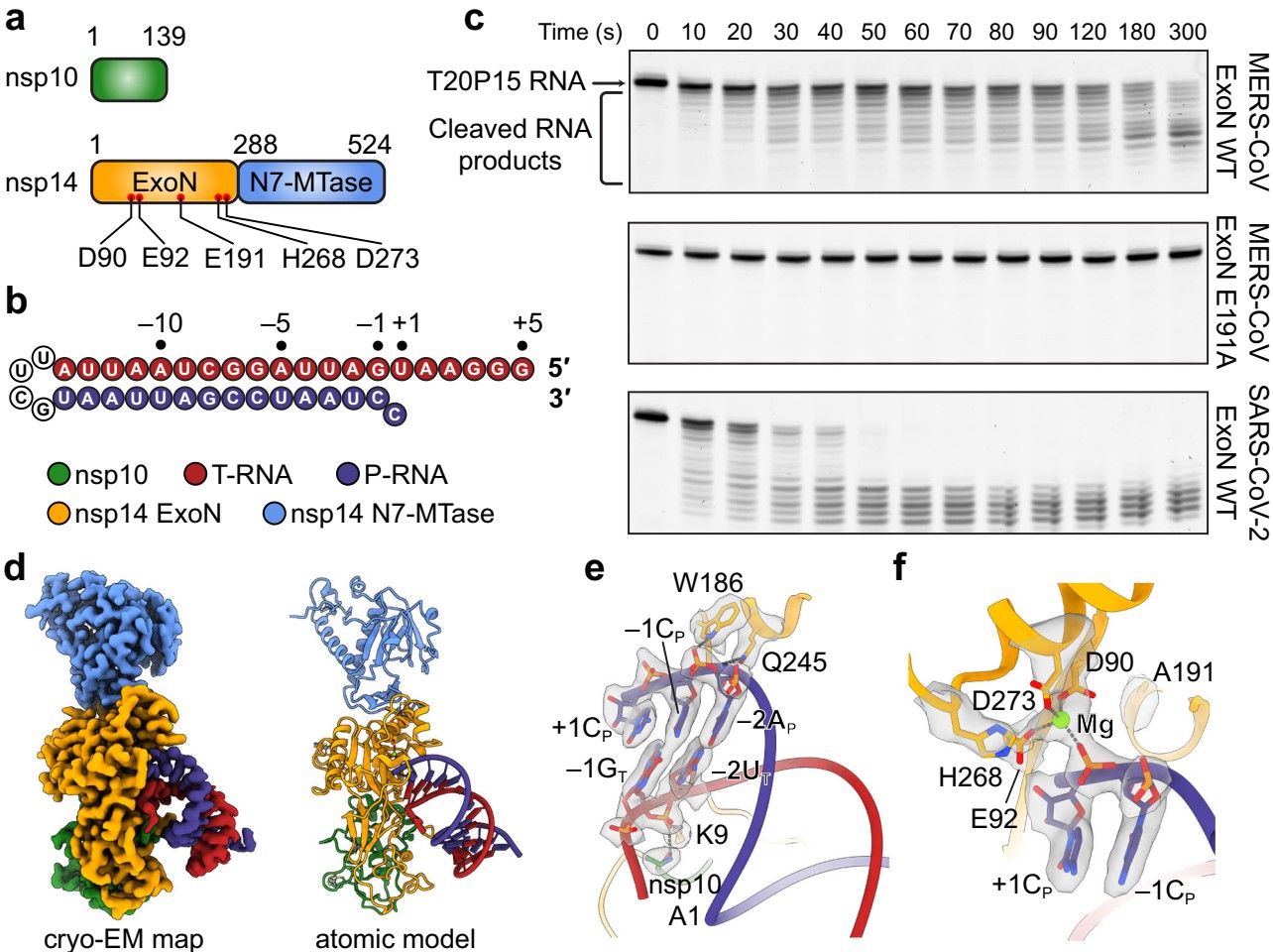

**Fig. 1 | Structure and catalytic activity of MERS-CoV ExoN complex. a** Domain organization of MERS-CoV nsp10 and nsp14. Domain boundaries are shown by residue number. Active site residues are indicated as red dots and labeled. **b** Sequence and numbering of the T20P15 RNA used in exoribonuclease assay and cryo-EM analysis. T-RNA and P-RNA regions are colored in red and blue, respectively, and connected by a UUCG tetraloop. **c** Exonucleolytic digestion of T20P15 RNA by MERS-CoV ExoN WT, MERS-CoV ExoN E191A mutant, or SARS-CoV-2 ExoN WT. Reactions were stopped at indicated time points, and RNA products were resolved by denaturing polyacrylamide gel electrophoresis (PAGE) and stained by SYBR-Gold. A representative result from three biological replicates is shown.

**d** Cryo-EM map and atomic model of MERS-CoV ExoN•T20P15 RNA complex. **e** Overview of the interaction between MERS-CoV ExoN and T20P15 RNA. Nucleotide residues in T-RNA and P-RNA are indicated with a subscript "T" or "P", respectively. Hydrogen bonds are shown as gray dashed lines. Nucleotides bound in the shallow RNA-binding pocket of ExoN and RNA backbone-interacting amino acid residues of ExoN are superimposed with their cryo-EM densities contoured at 7σ. **f** Active site conformation of the MERS-CoV ExoN•T20P15 RNA complex. Mg$^{2+}$ ions, green spheres. +1C$_P$, −1C$_P$, Mg$^{2+}$ ion, and two active site residues are superimposed with their cryo-EM densities contoured at 7σ. Source data are provided as a Source Data file.

two coronavirus ExoNs, we first compared the digestion of MERS-CoV ExoN and SARS-CoV-2 ExoN on the T20P15 RNA. Our result shows that while MERS-CoV ExoN is highly active in cleaving the RNA substrate, its digestion rate is markedly lower than that of SARS-CoV-2 ExoN (Fig. 1c). For a more thorough interrogation of the enzymatic kinetics, we next measured the exonucleolytic digestion of a fluorescence-labeled hairpin RNA bearing a 3′-end mismatched cytidine (hereafter referred to as T20P14-misC. Full sequence of the RNA is listed in Supplementary Table 2) by the two coronavirus ExoNs, respectively, and compared their Michaelis-Menten kinetics. While the Michaelis constant ($K_M$) values are comparable (4.919 μM for MERS-CoV ExoN and 4.980 μM for SARS-CoV-2 ExoN), indicating similar RNA-binding affinity, between the two ExoNs, the turnover number and overall catalytic efficiency of MERS-CoV ExoN ($k_{cat}$ = 46.90 min$^{-1}$, $k_{cat}/K_M$ = 9.534 μM$^{-1}$•min$^{-1}$) are greatly lower than those of SARS-CoV-2 ExoN ($k_{cat}$ = 309.4 min$^{-1}$, $k_{cat}/K_M$ = 62.13 μM$^{-1}$•min$^{-1}$) (Fig. 2e). Furthermore, to examine the effect of Mg$^{2+}$ concentration on the catalytic activities of MERS-CoV and SARS-CoV-2 ExoNs, we determined the digestion rate constants ($k$) of the two coronavirus ExoNs across a

range of Mg$^{2+}$ concentrations. While both ExoNs experience a gradual boost of their catalytic activity as Mg$^{2+}$ concentration increases from 1 mM to 16 mM, the digestion rate constant of SARS-CoV-2 ExoN is substantially higher than that of MERS-CoV ExoN at each of the tested Mg$^{2+}$ concentrations (Fig. 2f). Therefore, our result indicates that the considerably lower catalytic activity of MERS-CoV ExoN than that of SARS-CoV-2 ExoN is not because of its preference to a different Mg$^{2+}$ concentration, but rather likely stems from the suboptimal conformation of its catalytic residue H268 (Fig. 2b). Taken together, our structural analyses and biochemical characterizations demonstrate that the active site of ExoN from MERS-CoV predominantly adopts a conformation that is less readily activated by the binding of RNA substrate, explaining the lower catalytic activity compared with SARS-CoV-2 ExoN.

**Nucleotides recognition by ExoN**

While the difference in active site conformation between MERS-CoV and SARS-CoV-2 ExoNs leads to a substantial difference in their catalytic activities, it does not greatly alter the recognition pattern of

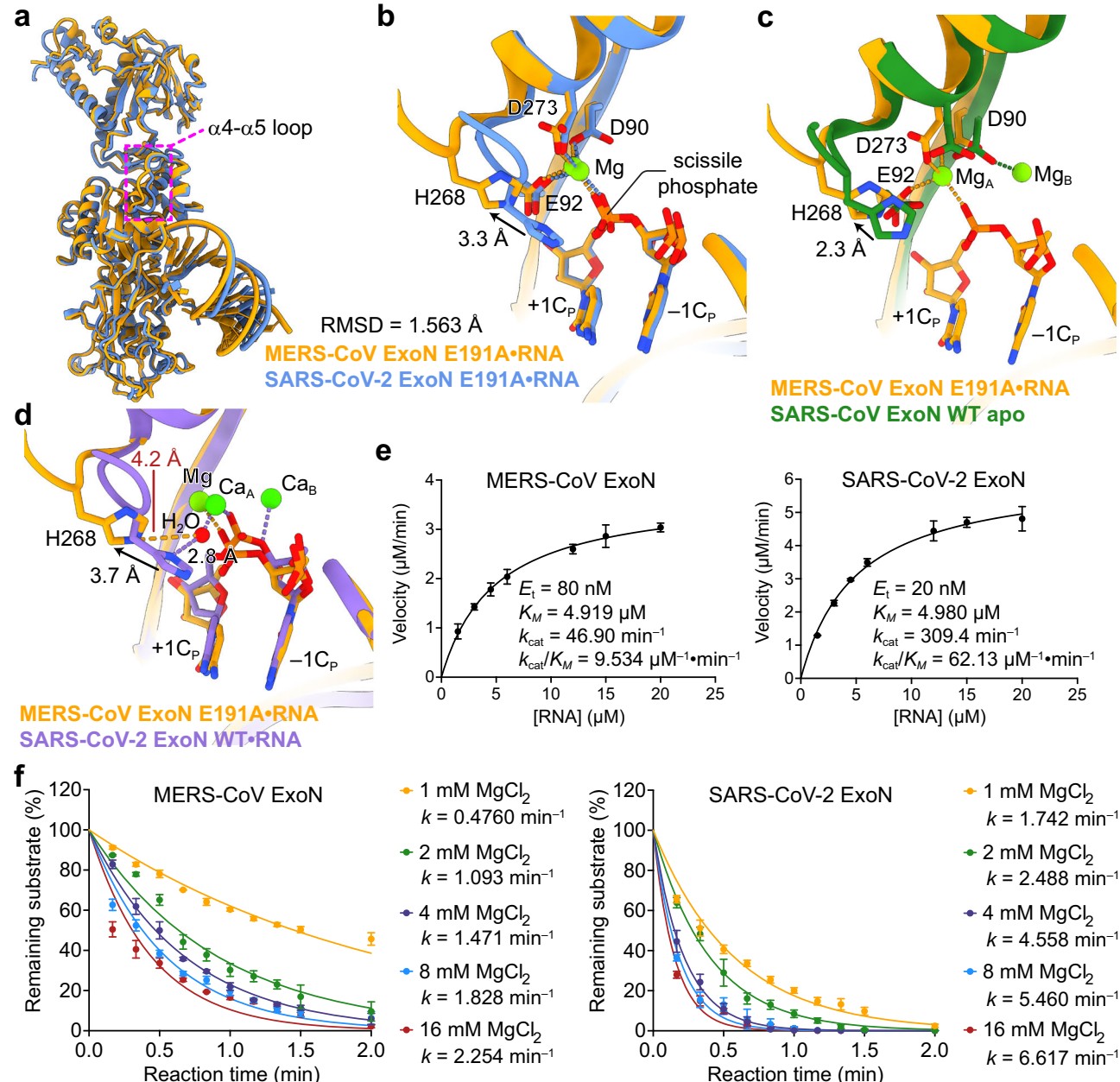

**Fig. 2 | Structural and catalytic divergence of MERS-CoV and SARS-CoV-2 ExoN complex. a** Superimposition of MERS-CoV ExoN E191A•T20P15 RNA complex (colored in orange) and the SARS-CoV-2 ExoN E191A•RNA complex (PDB ID 7N0C, colored in blue). The root-mean-square deviation (RMSD) between the two structures is indicated. The conformational difference of the α4-α5 loop, which harbors the ExoN catalytic residue H268, between the two ExoNs is highlighted in a dashed magenta box. **b** A close-up view of the ExoN active site superimposition between the two ExoN•RNA complexes illustrates a 3.3 Å shift of H268 away from the scissile phosphate in the MERS-CoV ExoN•RNA complex. **c** Superimposition of the ExoN active site between MERS-CoV ExoN E191A•T20P15 RNA complex and the RNA-free SARS-CoV ExoN complex (PDB ID 5C8U). **d** Superimposition of the ExoN active site between MERS-CoV ExoN E191A•T20P15 RNA complex (colored in orange) and the SARS-CoV-2 ExoN WT•RNA complex (PDB ID 7N0B, colored in medium purple). The

distances between the catalytic water molecule (red sphere) and H268 of MERS-CoV ExoN or H268 of SARS-CoV-2 ExoN are indicated. **e** Michaelis-Menten kinetics of MERS-CoV and SARS-CoV-2 ExoNs. Michaelis-Menten parameters were obtained by nonlinear regression. Values of $k_{cat}$ were obtained by dividing the $V_{max}$ values by the concentration of ExoN ($E_t$). **f** Impact of $Mg^{2+}$ concentration on the RNA digestion activity of MERS-CoV and SARS-CoV-2 ExoNs. Exonucleolytic digestion of T20P15 RNA substrate by MERS-CoV or SARS-CoV-2 ExoN was stopped at indicated time points, and RNA products were resolved by denaturing PAGE and stained by SYBR-Gold. Percentage of substrate RNA remaining at each time point was quantified using Bio-Rad Image Lab from three independent experiments and is shown as mean ± SEM. Rate constant ($k$) values are determined using the One-phase decay model in GraphPad Prism. Source data are provided as a Source Data file.

the 3′-end cytidine by MERS-CoV ExoN in comparison to SARS-CoV-2 ExoN. Specifically, MERS-CoV ExoN establishes multiple ribose- and base-specific hydrogen bonds with +1C$_P$ (Fig. 3a), which contrasts with the interactions of the ExoN with RNA phosphate backbones at −1 and −2 positions (Fig. 1e). These extensive interactions between ExoN and the +1C$_P$ nucleotide, together with the $Mg^{2+}$ ion-mediated

coordination of the scissile phosphate, properly positions the 3′-end nucleotide for excision.

To illuminate how coronavirus ExoN recognizes and excises different types of nucleotides at the 3′ end of a dsRNA substrate, which is important for its functions both as an RNA synthesis proofreader and a dsRNA-degrading immune suppressor, we also determined the cryo-

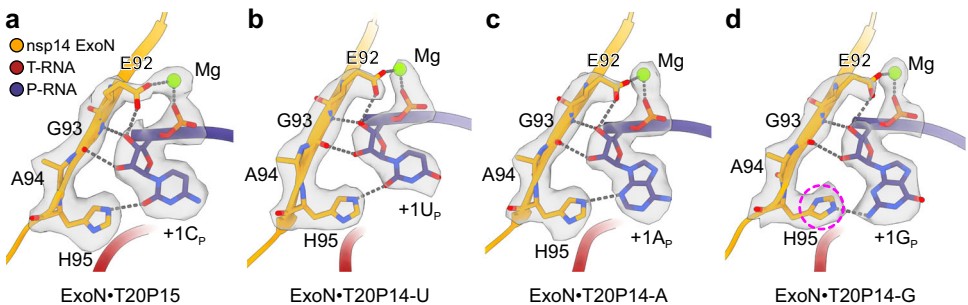

**Fig. 3 | Nucleotide recognition by MERS-CoV ExoN complex. a** Interactions between MERS-CoV ExoN and RNA 3′-end cytidine. **b** Interactions between MERS-CoV ExoN and RNA 3′-end uridine. **c** Interactions between MERS-CoV ExoN and RNA 3′-end adenosine. **d** Interactions between MERS-CoV ExoN and RNA 3′-end guanosine. Mg²⁺ ion, green sphere. Hydrogen bonds are shown as gray dashed lines.

The RNA 3′-end nucleotides and their interacting residues in nsp14 are superimposed with their cryo-EM densities contoured at 7σ. The distinct side chain rotamer conformation of H95 in the ExoN•T20P14-G complex is highlighted in a dashed magenta circle.

EM structures of MERS-CoV ExoN E191A mutant in complex with a hairpin RNA bearing a 3′-end adenosine (designated T20P14-A), uridine (designated T20P14-U), or guanosine (designated T20P14-G), respectively (Full sequences of the RNAs are listed in Supplementary Table 2). The final cryo-EM maps for all three ExoN•RNA complexes were refined to 2.9 Å (Supplementary Fig. 2–5). Superimpositions of each of the three ExoN•RNA complexes with the above-described MERS-CoV ExoN•T20P15 complex show that varying the 3′-end nucleotide of the RNA substrate does not reshape the overall structure of the complex or alter the suboptimal conformation of the catalytic residue H268 (Supplementary Fig. 6). A more detailed comparison of these four MERS-CoV ExoN•RNA complexes with regards to their RNA-binding mechanisms reveals that the specific recognition of 2′- and 3′-OH groups is a shared feature among the four different 3′-end nucleotides (Fig. 3b–d). In addition, nsp14 H95, which is approximately co-planar with the 3′-end nucleotides, forms a hydrogen bond with each of the four nucleotides at the RNA substrate 3′ end (Fig. 3a–d). The ability of the histidine imidazole side chain to serve as either a hydrogen bond donor or acceptor and to adopt multiple rotamer conformations allows such versatile patterns of hydrogen bonding and facilitates the proper recognition of each of four standard nucleobases.

## Substrate preference of ExoN

To quantitatively assess the preference of ExoN towards different 3′-end nucleotides of the RNA substrate, we performed RNA cleavage assays and determined the rate constants ($k$) of the digestion of a series of dsRNA substrates bearing a 3′-end mismatched cytidine (designated T20P14-misC), base-paired cytidine (designated T20P14-pairC), base-paired adenosine (T20P14-A), base-paired uridine (T20P14-U), or base-paired guanosine (T20P14-G) by MERS-CoV ExoN and SARS-CoV-2 ExoN, respectively (Fig. 4a,c and Supplementary Fig. 7a,c). Our results show that while the ExoNs from both coronaviruses are highly active on all five RNA substrates, which is consistent with our structural observations (Fig. 3) and a reflection of the broad-spectrum activity of coronavirus ExoNs on RNAs with diverse sequences, the rates of digestion on different RNAs vary significantly. Both MERS-CoV and SARS-CoV-2 ExoNs most readily digest T20P14-misC, followed by T20P14-A and T20P14-U. By contrast, the digestion of T20P14-pairC is significantly slower than the digestion of the above three RNA substrates, whereas T20P14-G exhibits the lowest rate constant of digestion by the two coronavirus ExoNs (Fig. 4a,b and Supplementary Fig. 7a,b). Consistent with the results of our RNA cleavage assays, fluorescence polarization assays performed using the same set of dsRNAs and ExoN E191A mutants from MERS-CoV and SARS-CoV-2 reveal the same order of nucleotide preference. Both ExoNs exhibit the highest binding affinity to T20P14-misC and the lowest binding affinity to T20P14-G. The ExoN-binding affinities of T20P14-A, T20P14-U, and

T20P14-pairC fall within an intermediate range on the spectrum (Fig. 4c and Supplementary Fig. 7c).

Such an order of preference of ExoN to different nucleotides is rooted in how ExoN interacts with the 3′ end nucleotide of a dsRNA substrate. As revealed by the SARS-CoV-2[23] and our MERS-CoV ExoN•RNA complex structures (Fig. 3), a separation of the 3′-end base pair or mismatched pair is required for an optimal binding of the RNA substrate by ExoN and correct positioning of the 3′-end nucleotide of the P-RNA strand at the ExoN catalytic center for excision. Therefore, the lower energetic barrier to break a mismatched U:C pair than an A:U or U:A base pair and the even more stable G:C or C:G base pair dictates the strongest binding of ExoN to and hence the highest cleavage activity on the T20P14-misC RNA (Fig. 4a–c and Supplementary Fig. 7a–c). Notably, in the cryo-EM structure of the ExoN•T20P14-G complex, H95 adopts a rotamer conformation that is different from that observed in the other three ExoN•RNA complexes (Fig. 3). While this different rotamer conformation of H95 allows it to establish a hydrogen bond with the guanine base (Fig. 3d), it is much less favorable (7.78% probability, as reported by the Rotamer Analysis tool in Coot[28]) than the rotamer conformation adopted when H95 interacts with the 3′-end cytidine, uridine, or adenosine (51.29% probability on average across the three structures) (Fig. 3a–c). In addition, grafting H95 in the rotamer conformation as observed in the ExoN•T20P15 complex to the ExoN•T20P14-G complex leads to a clash between the Cε1-Hε1 group of H95 and N2-amine group of the 3′-end guanine base (Supplementary Fig. 6e). Therefore, the less preferred recognition of its guanine base by nsp14 H95 and the higher energy penalty to break the base pair formed by the 3′-end guanosine together result in the lowest activity of ExoN on the T20P14-G RNA (Fig. 4a–c and Supplementary Fig. 7a–c).

## Structural determinants of broad substrate specificity of ExoN

Our collection of cryo-EM structures and biochemical characterizations of ExoN's binding to and exonucleolytic activity on different RNA substrates strongly indicate that nsp14 H95 is an important structural determinant of the broad-spectrum activity of coronavirus ExoN on RNA substrates with diverse sequences. To further interrogate the role of H95 in ExoN-mediated RNA digestion, we measured the digestion of the above-described T20P14 series of RNAs by either MERS-CoV or SARS-CoV-2 ExoN containing an nsp14 H95A mutation (hereafter referred to as ExoN H95A) (Fig. 4d–f and Supplementary Fig. 7d–f). Compared with WT ExoN, ExoN H95A mutants from both MERS-CoV and SARS-CoV-2 show significantly weakened cleavage activities on all five RNA substrates (Fig. 4d–f and Supplementary Fig. 7d–f). Such reduced cleavage of this diverse range of RNA substrates by the ExoN H95A mutant is likely due to a loss of the favorable hydrogen bond formed between H95 and each type of nucleotide (Fig. 3) and

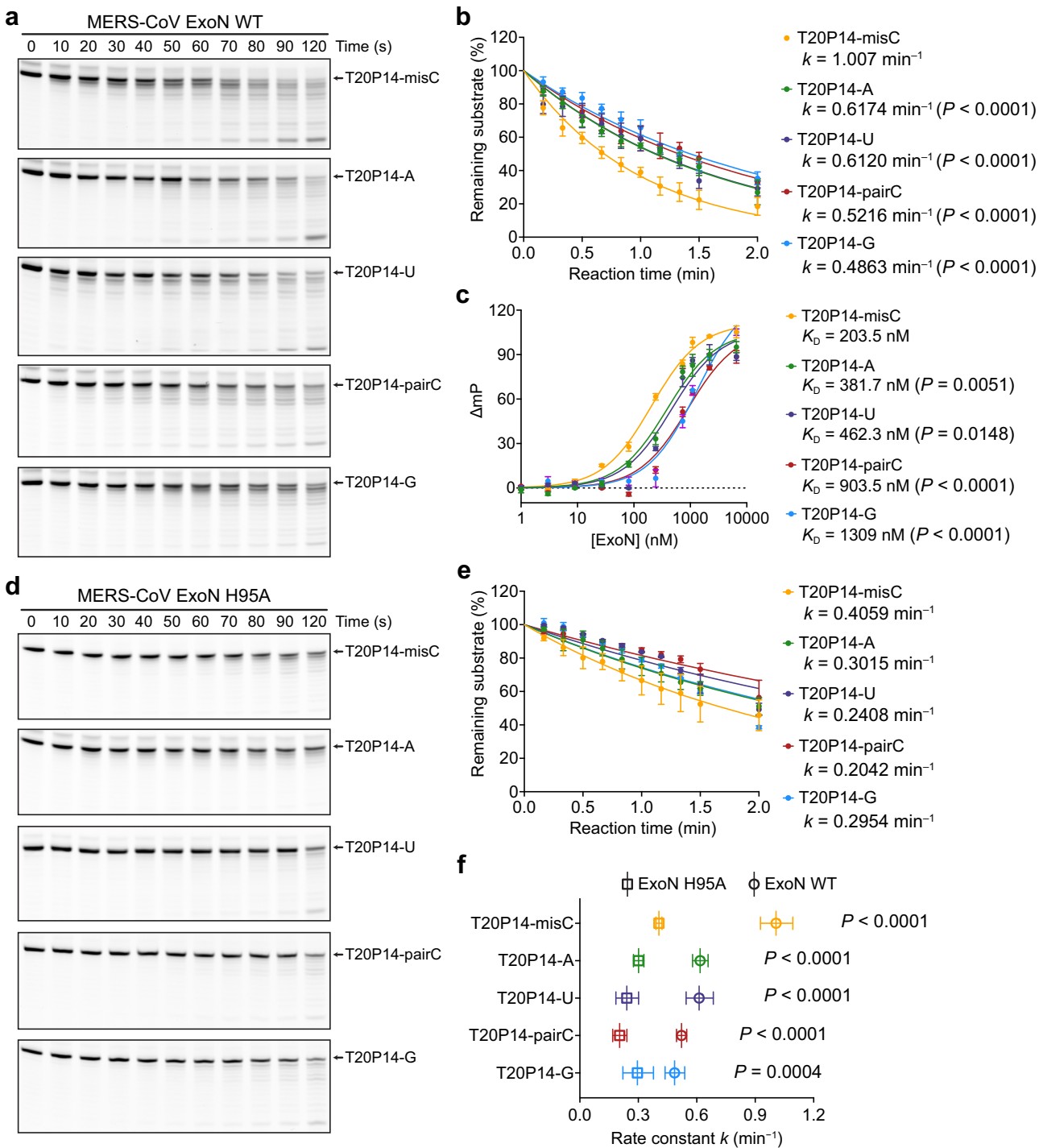

**Fig. 4 | Substrate preference of MERS-CoV ExoN complex. a** Exonucleolytic digestion of RNA substrates bearing different 3′-end nucleotides by MERS-CoV ExoN WT. The reactions were stopped at indicated time points, and RNA products were resolved by denaturing PAGE and visualized by FAM (fluorescein) imaging. A representative result from three biological replicates is shown. **b** Percentages of substrate RNAs remaining shown in **a** were quantified using Bio-Rad Image Lab from three independent experiments and are shown as mean ± SEM. The results were plotted in GraphPad Prism using the One-phase decay model. Rate constant ($k$) values are indicated. Statistical analyses were performed using the two-sided extra sum-of-squares F test[57]. $P$ values for the comparison of rate constants between T20P14-misC and each of the other four RNAs are indicated. **c** Fluorescence polarization analysis of the binding between MERS-CoV ExoN complex and different RNA substrates. Each data point represents the mean of six biological replicates ± SEM. Dissociation constant ($K_D$) values are indicated.

Statistical analyses were performed using the two-sided extra sum-of-squares F test. $P$ values for the comparison of $K_D$ between T20P14-misC and each of the other four RNAs are indicated. **d** Exonucleolytic digestion of RNA substrates bearing different 3′-end nucleotides by MERS-CoV ExoN H95A mutant. A representative result from three biological replicates is shown. **e** Percentages of substrate RNAs remaining shown in **d** were quantified using Bio-Rad Image Lab from three independent experiments and are shown as mean ± SEM. The results were plotted in GraphPad Prism using the One-phase decay model. Rate constant ($k$) values are indicated. **f** Comparison of the rate constants of RNA digestion by MERS-CoV ExoN WT and H95A mutant. Data are shown as best-fit rate constant values ± 95% confidence interval (CI) of three biological replicates determined from curve fitting in **b** and **e**. Statistical analyses were performed using the two-sided extra sum-of-squares F test. $P$ values are indicated. Source data are provided as a Source Data file.

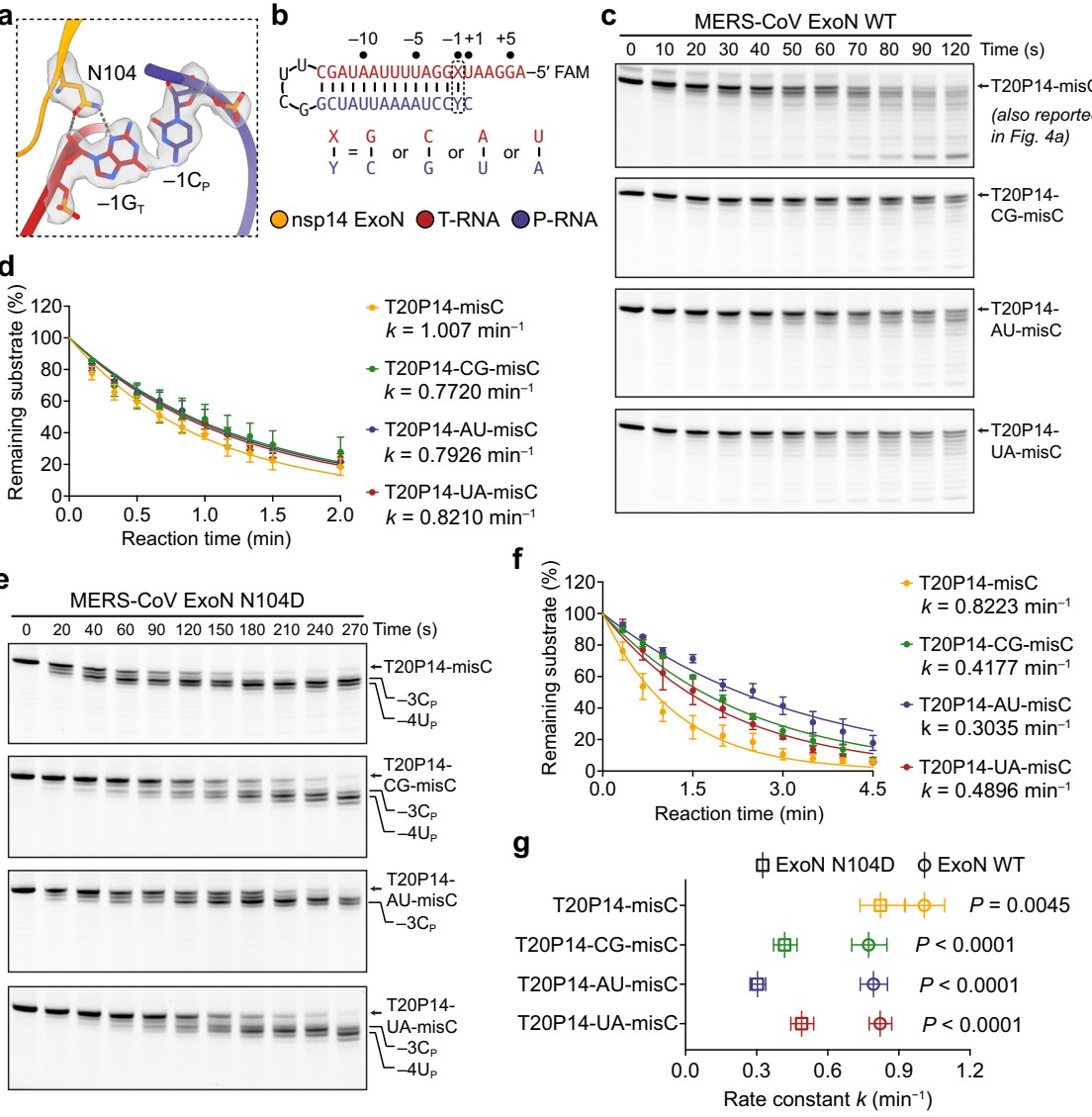

**Fig. 5 | Nsp14 N104 facilitates the sequence-independent digestion of dsRNA substrates by MERS-CoV ExoN. a** Interactions between MERS-CoV ExoN and T20P15 RNA substrate mediated by nsp14 N104. Hydrogen bonds are shown as dashed gray lines. Nsp14 N104 and the G:C base pair at the RNA −1 position are superimposed with their cryo-EM densities contoured at 7σ. **b** Sequence and numbering of the T20P14-misC RNA and its three variants used in the exoribonuclease assay. **c** Exonucleolytic digestion of RNA substrates bearing different base pairs at the −1 position by MERS-CoV ExoN WT. The reactions were stopped at indicated time points, and RNA products were resolved by denaturing PAGE and visualized by FAM (fluorescein) imaging. A representative result from three biological replicates is shown. **d** Percentages of substrate RNAs remaining shown in **c** were quantified using Bio-Rad Image Lab from three independent experiments and are shown as mean ± SEM. The results were plotted in GraphPad Prism using the One-phase decay model. Rate constant ($k$) values are indicated.
**e** Exonucleolytic digestion of RNA substrates bearing different base pairs at the −1 position by MERS-CoV ExoN N104D mutant. A representative result from three biological replicates is shown. **f** Percentages of substrate RNAs remaining shown in **e** were quantified using Bio-Rad Image Lab from three independent experiments and are shown as mean ± SEM. The results were plotted in GraphPad Prism using the One-phase decay model. Rate constant ($k$) values are indicated. **g** Comparison of the rate constants of RNA digestion by MERS-CoV ExoN WT and N104D mutant. Data are shown as best-fit rate constant values ± 95% CI of three biological replicates determined from curve fitting in **d** and **f**. Statistical analyses were performed using the two-sided extra sum-of-squares F test. *P* values are indicated. Source data are provided as a Source Data file.

corroborates the critical role of H95 in rendering ExoN broadly active in excising different nucleotides from RNA substrates.

In addition to H95, nsp14 N104 is the only other ExoN residue that makes specific interactions with the ribose and nucleobase moieties in the RNA substrate. In all four ExoN•RNA complex structures reported here, the amide side chain of N104 forms two hydrogen bonds with the 2′-OH and N3-imine of −1$G_T$, respectively (Fig. 5a). In addition to the conformation built in our ExoN•RNA complex structures, the N104 amide side chain can presumably adopt a flipped conformation while also establishing two hydrogen bonds with −1$G_T$ (Supplementary Fig. 8a). The two alternative conformations of N104 are difficult to be

distinguished from each other at the resolutions commonly achieved by cryo-EM and are likely simultaneously present in the ExoN•RNA complex particles used to reconstruct the cryo-EM maps.

To understand how N104 accommodates other nucleotides at the −1 position in T-RNA, we substituted −1$G_T$ in our ExoN•T20P15 complex structure with a cytidine, uridine, and adenosine, respectively. The structural modeling suggests that the two hydrogen bonds mediated by N104 are maintained when each of the four nucleotides is present at the −1 position of T-RNA (Supplementary Fig. 8b–d). To validate our structural observations and examine how different nucleotide sequences at the −1 position affect the RNA cleavage by coronavirus

ExoN, we replaced the −1 $G_T$:$C_P$ base pair in the T20P14-misC RNA with a $C_T$:$G_P$, $A_T$:$U_P$, or $U_T$:$A_P$ base pair (designated T20P14-CG-misC, T20P14-AU-misC, or T20P14-UA-misC, respectively) (Fig. 5b) and determined the exonucleolytic digestion rate constant of these RNAs by MERS-CoV or SARS-CoV-2 ExoN, respectively. Our results show that both coronavirus ExoNs exhibit robust cleavage activities on all four RNAs (Fig. 5c,d and Supplementary Fig. 9a,b). In addition, the more versatile hydrogen bonding patterns between −1$G_T$ and either side chain conformation of N104 (Fig. 5a and Supplementary Fig. 8a) likely result in a moderately higher cleavage activity on the original T20P14-misC RNA containing a −1$G_T$ than on the other three RNAs (Fig. 5d and Supplementary Fig. 9b), in which the −1$C_T$, −1$A_T$, or −1$U_T$ nucleotide restricts the side chain of N104 to one conformation for favorable hydrogen bonding interactions (Supplementary Fig. 8b–d).

To further assess the contribution of nsp14 N104 to the cleavage of RNAs with diverse sequences by coronavirus ExoN, we quantified the exonucleolytic digestion activities of the four different variants of T20P14-misC RNAs by MERS-CoV or SARS-CoV-2 ExoN containing an nsp14 N104D mutation (hereafter referred to as ExoN N104D mutant) (Fig. 5e, f and Supplementary Fig. 9c, d) and compared the digestion rate constants between ExoN WT and ExoN N104D. We found that N104D mutation of ExoN leads to significantly reduced cleavage activity on all four RNAs (Fig. 5g and Supplementary Fig. 9e). In addition, such N104D-mediated disruption of RNA cleavage is more pronounced in RNAs containing −1$C_T$, −1$A_T$, or −1$U_T$ than in the RNA containing −1$G_T$ (Fig. 5g and Supplementary Fig. 9e). To provide a structural explanation to this uneven reduction of cleavage activity towards different RNAs, we modeled the interactions between a mutated D104 and −1$G_T$, −1$C_T$, −1$A_T$, or −1$U_T$ of the RNA substrates, respectively. Our structural modeling shows that while D104 can potentially form two hydrogen bonds with −1$G_T$, this mutated ExoN residue disfavors cytosine, adenine, or uracil base and likely destabilizes the binding of the RNA substrate containing −1$C_T$, −1$A_T$, or −1$U_T$ (Supplementary Fig. 8e–h). Consistent with the substantially impaired cleavage of RNA containing −1$A_T$ or −1$U_T$ by ExoN N104D, there is considerable stalling of ExoN N104D-mediated RNA cleavage at −3$C_P$ and −4$U_P$ when −4$A_T$ and −5$U_T$ in the original RNA substrates has now moved to the −1 position of T-RNA (Fig. 5e and Supplementary Fig. 9c). Taken together, our results strongly suggest that nsp14 N104 dictates the proper recognition of different nucleotides at the −1 position of T-RNA and active digestion of a wide range of RNA substrates with diverse sequences by coronavirus ExoNs.

## Discussion

Proofreading ExoN is present in all coronaviruses known to date and plays key roles in viral survival and fitness through a multi-thronged mechanism, including improving the fidelity of viral RNA synthesis and mediating the evasion of host IFN-dependent antiviral responses. Because of its essential functions in the replication and life cycle of all coronaviruses, ExoN represents a clinically significant drug target for the development of pan-coronavirus antiviral therapeutics[3,5,21,29]. However, current knowledge of ExoN derives from the studies of a limited range of coronavirus species. In particular, the structural basis and molecular details of RNA substrate recognition by ExoN have only been characterized for ExoN from SARS-CoV-2[23,30], leaving a major gap in our understanding of the evolutionary conservation and divergence of the structure, catalytic activity, and substrate specificity of ExoN across different coronaviruses. Recent discoveries of novel bat and mink coronaviruses that are closely related to the highly pathogenic MERS-CoV and readily infect human cells[31–38] spark widespread attention and concern about potential outbreaks of new zoonotic MERS-CoV-like coronaviruses and highlight the importance of detailed characterizations of essential viral components of MERS-CoV and related coronaviruses in the merbecovirus subgenus.

In this study, we determined the first set of high-resolution cryo-EM structures of ExoN from a merbecovirus, MERS-CoV, in complex with a series of dsRNA substrates containing various 3′-end nucleotides. Comprehensive comparisons of the structures, catalytic activities, and substrate preferences between MERS-CoV and SARS-CoV-2 ExoNs reveal that, while the two coronavirus ExoNs greatly differ in their catalytic efficiency, likely due to a divergence of their active site conformation, they exhibit a similar order of preferences to different nucleotides at key positions of the dsRNA substrates. In general, the two coronavirus ExoNs readily degrade dsRNAs regardless of their nucleotide sequences. Such broad-spectrum digestion activity of coronavirus ExoNs on dsRNAs with different sequences is primarily conferred by H95 and N104 of nsp14, which can properly recognize all four nucleotides due to their versatile hydrogen bonding patterns. The broad substrate specificity of coronavirus ExoNs could be critical for their roles not only in degrading viral dsRNAs in a sequence-independent manner but also in removing different nucleotides mis-incorporated into the nascent viral RNAs by the error-prone viral RdRp.

Our findings suggest that ExoNs from different coronaviruses can possess distinct local structures and diverge in their enzymatic activities or biological functions despite their relatively conserved overall architecture and catalytic mechanism, underscoring the necessity of future comparative studies of ExoN or other essential viral enzymes from evolutionarily distant coronavirus species. Moreover, the cryo-EM structures depicting the atomic details of RNA substrate binding and the detailed substrate preference of MERS-CoV and SARS-CoV-2 ExoNs elucidated in this study can provide insights into the design of potent inhibitors targeting ExoNs from multiple current and potentially future coronaviruses.

## Methods

### Protein expression and purification

The genes of MERS-CoV nsp10, nsp14, and SARS-CoV-2 nsp10 were chemically synthesized with codon optimization for expression in *Escherichia coli* (*E. coli*) (Integrated DNA Technologies). The gene of SARS-CoV-2 nsp14 was requested from Addgene. The MERS-CoV and SARS-CoV-2 nsp14 genes were cloned into a pETDuet-1 vector with an N-terminal His$_6$-Smt3 tag between the NcoI and HindIII restrictive sites. The MERS-CoV and SARS-CoV-2 nsp10 genes were cloned into a pETDuet-1 vector between the NdeI and XhoI restrictive sites. E191A, H95A, or N104D mutation of nsp14 was introduced by site-directed mutagenesis. The mutation was confirmed by Sanger sequencing. Plasmids are available upon request.

All proteins were overexpressed in *E. coli* BL21 Star (DE3) (ThermoFisher Scientific) at 17 °C for 18 h. Cells were resuspended in buffer A (50 mM 4-(2-hydroxyethyl)-1-piperazine ethanesulfonic acid (HEPES), pH 7.5, 200 mM NaCl, 5% glycerol, 1 mM β-mercaptoethanol (β-ME), 20 mM imidazole) and lysed using a sonicator (QSonica). The cell lysate was cleared by centrifugation at 19,500 rpm using a JA-25.50 rotor (Beckman Coulter) for 1 h at 4 °C. The clarified cell lysate was loaded onto a HisTrap HP affinity chromatography column (Cytiva Life Sciences) and eluted through a linear gradient from 100% buffer A to 40% buffer A mixed with 60% buffer B (50 mM HEPES, pH 7.5, 200 mM NaCl, 5% glycerol, 1 mM β-ME, 500 mM imidazole). Eluted protein samples were loaded onto a HiTrap Heparin HP column (Cytiva Life Sciences) and eluted by buffer C (20 mM HEPES, pH 7.0, 1 M NaCl, 2 mM β-ME), followed by incubation with Ulp1 protease to remove the N-terminal His$_6$-Smt3 tag. All proteins were subsequently purified by size-exclusion chromatography (SEC) on a HiLoad 16/600 Superdex 200 pg column (Cytiva Life Sciences) in buffer D (20 mM HEPES, pH 7.5, 150 mM NaCl, 5 mM MgCl$_2$, 1 mM Tris(2-carboxyethyl) phosphine hydrochloride (TCEP-HCl)). ExoN complex was assembled by mixing nsp10 and nsp14 in a 1:1 molar ratio at room temperature for 30 min.

## In vitro transcription and RNA purification

The DNA template for in vitro transcription of T45P24CG RNA was generated by annealing two DNA oligonucleotides with complementary sequences (5′-CAC TAA TAC GAC TCA CTA TAG GGA ATG ATT AGG CTA ATT ATT CGT AAT TAG CCT AAT CC-3′ and 5′-[G$_m$][G$_m$]A TTA GGC TAA TTA CGA ATA ATT AGC CTA ATC ATT CCC TAT AGT GAG TCG TAT TAG TG-3′) in a 1:1 molar ratio. T7 RNA polymerase (RNAP) φ6.5 promoter sequence in the non-template DNA strands is underlined. Two nucleotides denoted [G$_m$] on the 5′ end of the template DNA strand are 2′-O-methylated to improve the 3′-end homogeneity of the RNA transcripts[39].

In vitro transcription reaction (5 mL) was assembled using 2 nmol annealed DNA template, 500 µg T7 RNAP and 200 µl 25× RNAsecure RNase inactivation reagent (ThermoFisher Scientific) in 1× reaction buffer (80 mM HEPES, pH 7.5, 24 mM MgCl$_2$, 40 mM DTT, 2 mM spermidine, 4 mM of each NTP) and incubated at 37 °C for 2 h. The reaction mixture was centrifuged at 4000 × g for 10 min at 4 °C to remove pyrophosphate precipitate and subsequently quenched by adding ethylenediaminetetraacetic acid (EDTA) to a final concentration of 50 mM. The RNA transcripts were extracted with phenol:chloroform:isoamyl alcohol (25:24:1) (ThermoFisher Scientific) three times, followed by purification on a Sephadex G-25 PD-10 desalting column (Cytiva Life Sciences) and a HiLoad 16/600 Superdex 200 pg SEC column in buffer E (10 mM HEPES, pH 7.0, 50 mM NaCl).

## Cleavage of T20P15 RNA by MERS-CoV and SARS-CoV-2 ExoNs

Purified T20P15 RNA at a final concentration of 3 µM was incubated with 80 nM of MERS-CoV nsp10•nsp14 WT complex, MERS-CoV nsp10•nsp14 E191A mutant complex, or SARS-CoV-2 nsp10•nsp14 E191A mutant complex at 37 °C in reaction buffer (25 mM HEPES, pH 7.5, 50 mM NaCl, 4 mM MgCl$_2$, and 1 mM TCEP). To determine the dependence of MERS-CoV and SARS-CoV-2 ExoNs' activities on Mg$^{2+}$ concentration, the ExoN•T20P15 RNA mixtures were incubated at 37 °C in reaction buffer containing 1 mM, 2 mM, 4 mM, 8 mM, or 16 mM MgCl$_2$, respectively. The reactions were stopped at different time points as indicated in the figures, respectively, by adding an equal volume of 2 × TBE-Urea sample buffer (ThermoFisher Scientific) supplemented with 50 mM EDTA and heating at 75 °C for 5 min. The cleavage products were resolved on denaturing 18% polyacrylamide gels and stained by SYBR Gold (ThermoFisher Scientific) on a Chemi-Doc MP imager (Bio-Rad). The RNA band corresponding to the substrate RNA at each reaction time point was quantified using Image Lab Software Suite (Bio-Rad). Percentages of substrate RNAs remaining were plotted against their respective reaction times in GraphPad Prism. The results were subjected to curve-fitting using the One-phase decay model to determine the rate constant ($k$) of RNA digestion for each reaction.

## MERS-CoV ExoN•RNA complexes assembly

The four MERS-CoV ExoN•RNA complexes in this study were reconstituted by mixing MERS-CoV nsp10-nsp14 E191A mutant complex with RNA in a 1:2 molar ratio and incubating the mixture at 30 °C for 30 min in buffer F (25 mM HEPES, pH 7.5, 50 mM NaCl, 4 mM MgCl$_2$, and 1 mM TCEP). The assembled MERS-CoV ExoN•RNA complexes were purified using a Superdex 200 Increase 10/300 GL SEC column (Cytiva Life Sciences) in buffer F. The chromatography fractions corresponding to the ExoN•RNA complexes were collected for subsequent single-particle cryo-EM analysis.

## Cryo-EM sample preparation and data acquisition

Purified MERS-CoV ExoN•RNA complexes (A260 = 3) were mixed with 8 mM of 3-([3-Cholamidopropyl]dimethylammonio)-2-hydroxy-1-propanesulfonate (CHAPSO) immediately before grid preparation. 3.5 µl of each complex was applied to freshly glow-discharged Quantifoil 300 mesh holey carbon grids with R1.2/1.3 hole pattern (Electron Microscopy Sciences). Grids were blotted for 5 s at 22 °C under 100% relative humidity and plunge-frozen in liquid nitrogen-cooled liquid ethane. The cryo-EM dataset for the ExoN•T20P15 complex, which comprises 5,817 movies, was collected on a Titan Krios electron microscope (ThermoFisher Scientific) operated at 300 kV equipped with a BioQuantum K3 detector (Gatan, Inc.) at the Stanford-SLAC Cryo-EM Center (S$^2$C$^2$). The movie frames were collected at a nominal magnification of 130,000 ×, corresponding to 0.653 Å per pixel, at a dose rate of 11 e$^-$ per physical pixel per second with a defocus range of −1.0 to −2.0 µm. The total exposure time for each movie is 2 s, thus resulting in a total accumulated dose of 51.59 e$^-$/Å2, which was fractionated into 40 frames. The cryo-EM datasets for ExoN•T20P14-U, ExoN•T20P14-A, and ExoN•T20P14-G complexes, which comprise 3,874 movies, 11,070 movies, and 9,020 movies, respectively, were collected on a Titan Krios electron microscope operated at 300 kV equipped with a Falcon 4i detector and Selectris X imaging filter (ThermoFisher Scientific) at the Beckman Center for cryo-EM at Johns Hopkins University. The movie frames were collected at a nominal magnification of 130,000 ×, corresponding to 0.93 Å per pixel, at a dose rate of 8.27 e$^-$ per physical pixel per second with a defocus range of −1.0 to −2.0 µm. The total exposure time for each movie is 4.5 s, thus resulting in a total accumulated dose of 43.03 e$^-$/Å2, which was fractionated into 40 frames. The statistics of cryo-EM data collection are summarized in Supplementary Table 1.

## Cryo-EM image processing

Dose-fractioned cryo-EM movies were imported into cryoSPARC[40] for patch-based motion correction and patch-based CTF estimation, followed by blob picking and Topaz picking[41]. Blob picking used a minimum particle diameter of 125 Å and a maximum particle diameter of 175 Å for all four cryo-EM datasets and yielded 618,359 particles for the ExoN•T20P15 complex dataset, 609,380 particles for the ExoN•T20P14-U complex dataset, 1,075,145 particles for the ExoN•T20P14-A complex dataset, and 1,050,042 particles for the ExoN•T20P14-G complex dataset, respectively. For Topaz picking, a model was generated by training for 30 epochs with a subset of ~5,000 particles from blob picking for each dataset. The generated Topaz picking model was then used for Topaz Extract job to pick particles from the entire dataset. Topaz picking yielded 687,267 particles for the ExoN•T20P15 complex dataset, 306,358 particles for the ExoN•T20P14-U complex dataset, 321,530 particles for the ExoN•T20P14-A complex dataset, and 682,219 particles for the ExoN•T20P14-G complex dataset, respectively. The picked particles were subjected to three rounds of 2D classifications to remove junk particles. Particles in good 2D classes were selected for the generation of multiple ab initio models, which were subsequently low-pass filtered to 20 Å and used as starting references for heterogeneous refinement in cryoSPARC or global 3D classification in RELION-5.0[42,43]. The particle stacks corresponding to good classes resulting from global 3D classification or heterogeneous refinement were subjected to iterative rounds of CTF refinement[44], Reference-based motion correction[45,46], and non-uniform refinement[47] to generate the final cryo-EM map.

To improve the map quality and interpretability of the MERS-CoV ExoN•T20P14-A and MERS-CoV ExoN•T20P14-G complexes, the final particle stacks corresponding to the two complexes were first refined with C2 symmetry imposed and subsequently subjected to symmetry expansion and particle subtraction to retain only the signal of the protomer A of the dimeric ExoN•RNA complexes, followed by masked local 3D refinement in cryoSPARC. The protomer A local refinement map from the ExoN•T20P14-A or ExoN•T20P14-G complex was further improved by density modification using Phenix Resolve[48] without supplying a structural model to avoid model bias. The overall map resolution was calculated based on the Fourier shell correlation (FSC) cutoff at 0.143 between two half-maps, after applying a soft mask to exclude the bulk solvent region. The maps were sharpened

automatically during non-uniform refinement or local refinement and post-processed using DeepEMhancer[49]. The raw maps, automatically sharpened maps, density-modified maps, and DeepEMhancer-processed maps were used as cross-references during model building. Local resolution estimation was calculated from the two half-maps in cryoSPARC and visualized in UCSF ChimeraX[50]. Histogram and direction FSC curves for cryo-EM maps were analyzed and generated by the Orientation Diagnosis tool[51,52] in cryoSPARC.

## Cryo-EM model building and refinement

The cryo-EM structure of SARS-CoV-2 nsp10•nsp14•RNA complex (PDB 7N0C) was first docked as a rigid body and then flexibility fitted[53] into each of the four MERS-CoV ExoN•RNA complex maps. The protein and RNA subunits in each complex were manually rebuilt in Coot[28]. The resolution and density features of the cryo-EM maps are sufficiently good for the unambiguous assignment of protein and RNA registers in each of the complexes. All atomic models were refined against their respective Resolve density-modified maps using Phenix real-space refinement[54] with secondary structure restraints, rotamer restraints, and Ramachandran restraints. The final structures were validated with MolProbity[55]. The Resolve density-modified map was used in validation to determine the model-map resolution for each structure. The statistics of cryo-EM refinement and validation were summarized in Supplementary Table 1. Molecular representations were prepared using UCSF ChimeraX.

## Michaelis-Menten kinetics of MERS-CoV and SARS-CoV-2 ExoNs

MERS-CoV nsp10•nsp14 WT complex at a final concentration of 80 nM or SARS-CoV-2 nsp10•nsp14 WT complex at a final concentration of 20 nM was incubated with FAM-T20P14-misC RNA at a final concentration of 1.5 μM, 3 μM, 4.5 μM, 6 μM, 12 μM, 15 μM, or 20 μM, respectively, at 37 °C in reaction buffer (25 mM HEPES, pH 7.5, 50 mM NaCl, 4 mM $MgCl_2$, and 1 mM TCEP). For reactions containing the RNA at 1.5–4.5 μM concentration, the reactions were stopped at 10 s, 20 s, 30 s, 40 s, 50 s, 60 s, 80 s, 100 s, and 120 s, respectively, by adding an equal volume of 2 × TBE-Urea sample buffer supplemented with 50 mM EDTA and heating at 75 °C for 5 min. For reactions containing the RNA at 6–20 μM concentration, the reactions were stopped at 30 s, 1 min, 1.5 min, 2 min, 2.5 min, 3 min, 4 min, and 5 min, respectively. The cleavage products were resolved on denaturing 18% polyacrylamide gels and visualized by fluorescent imaging on a ChemiDoc MP imager. The RNA band corresponding to the FMA-T20P14-misC substrate at each reaction time point was quantified using Image Lab Software Suite (Bio-Rad) and used to calculate the initial velocity ($v_0$) of substrate degradation at each RNA concentration. The results were plotted and subjected to curve-fitting in GraphPad Prism using the Michaelis-Menten model to determine the values of Michaelis constant ($K_M$), maximum velocity ($V_{max}$), and turnover number ($k_{cat}$) of MERS-CoV and SARS-CoV-2 ExoN complexes.

## Exoribonuclease assays

WT, H95A, or N104D mutant forms of MERS-CoV nsp10•nsp14 complex at a final concentration of 60 nM or SARS-CoV-2 nsp10•nsp14 complex at a final concentration of 20 nM was incubated with FAM-T20P14-misC, FAM-T20P14-U, FAM-T20P14-A, FAM-T20P14-pairC, FAM-T20P14-G, FAM-T20P14-CG-misC, FAM-T20P14-AU-misC, or FAM-T20P14-UA-misC with a final concentration of 3 μM at 37 °C in reaction buffer (25 mM HEPES, pH 7.5, 50 mM NaCl, 4 mM $MgCl_2$, and 1 mM TCEP). For reactions containing MERS-CoV nsp10•nsp14 N104D mutant, the reactions were stopped at 20 s, 40 s, 1 min, 1.5 min, 2 min, 2.5 min, 3 min, 3.5 min, 4 min, and 4.5 min, respectively, by adding an equal volume of 2 × TBE-Urea sample buffer supplemented with 50 mM EDTA and heating at 75 °C for 5 min. For the remaining reactions, they were stopped by 10 s, 20 s, 30 s, 40 s, 50 s, 60 s, 70 s, 80 s,

90 s, and 120 s, respectively. The cleavage products were resolved on denaturing 18% polyacrylamide gels and visualized by fluorescent imaging on a ChemiDoc MP imager. The RNA band corresponding to the substrate RNA at each reaction time point was quantified using Image Lab Software Suite (Bio-Rad). Percentages of substrate RNAs remaining were plotted against their respective reaction times in GraphPad Prism. The results were subjected to curve-fitting using the One-phase decay model to determine the rate constant ($k$) of RNA digestion for each reaction. Statistical analyses for comparing the best-fit rate constant ($k$) values between each group were performed using the extra sum-of-squares F test.

## Fluorescence polarization assays

FAM-labeled T20P14 series of RNAs at a final concentration of 6 nM were incubated with a 3x serial dilution series (ranging from 3 nM to 13.122 μM) of MERS-CoV or SARS-CoV-2 nsp10•nsp14 E191A mutant complex at room temperature in buffer F in a 96-well plate. Fluorescence polarization was measured on a Victor Nivo multimode microplate reader (Revvity) with the excitation and emission wavelengths of 480 nm and 530 nm, respectively. Changes in fluorescence polarization ($\Delta mP$) upon protein binding were plotted against ExoN E191A concentration in GraphPad Prism. The data were fitted using a custom "One site-specific binding with ligand depletion" model[56] [$Y = B_{max} \times (X - F \times Y/B_{max})/(K_D + X - F \times Y/B_{max})$, where $X$ is the total protein concentration, $F$ is the total fluorescence probe concentration, $Y$ is the change of fluorescence polarization from the RNA-only control group, $B_{max}$ is maximum binding in the same units as $Y$, and $K_D$ is the dissociation constant in the same unit as $X$] to determine the $K_D$ values and 95% confidence intervals. Statistical analyses for comparing the best-fit $K_D$ values between each group were performed using the extra sum-of-squares F test.

## Reporting summary

Further information on research design is available in the Nature Portfolio Reporting Summary linked to this article.

## Data availability

Atomic coordinates of five structures determined in this study have been deposited in the Protein Data Bank with accession codes 9YCK (MERS-CoV ExoN•T20P15 complex, monomeric form), 9YCL (MERS-CoV ExoN•T20P15 complex, dimeric form), 9YCM (MERS-CoV ExoN•T20P14-U complex), 9YCN (MERS-CoV ExoN•T20P14-A complex), and 9YCO (MERS-CoV ExoN•T20P14-G complex). The cryo-EM maps have been deposited in the Electron Microscopy Data Bank with accession numbers EMD-72775 (MERS-CoV ExoN•T20P15 complex, monomeric form), EMD-72776 (MERS-CoV ExoN•T20P15 complex, dimeric form), EMD-72777 (MERS-CoV ExoN•T20P14-U complex), EMD-72778 (MERS-CoV ExoN•T20P14-A complex), and EMD-72779 (MERS-CoV ExoN•T20P14-G complex). Source data are provided with this paper.

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

## Acknowledgements

We thank P. Juneja for support with cryo-EM sample screening at the Iowa State University cryo-EM facility, D. Sousa, D. Ding, and K. Cai at the Beckman Center for Cryo-EM at Johns Hopkins University, and G. Nye at the at the Stanford-SLAC Cryo-EM Center ($S^2C^2$) for support during the collection of cryo-EM datasets. The $S^2C^2$ is supported by a National Institutes of Health grant R24GM154186. This work was supported by a National Institutes of Health grant R35GM150607 to Y.Y. and a National Institutes of Health grant DP2AI177906 to C.L. and an award from the Searle Scholars Program SSP-2024-106 to C.L.

## Author contributions

C.L. and Y.Y. conceived and designed the experiments. Y.L. and X.C. performed protein and RNA purifications. Y.L., X.C., and L.M.R. performed biochemical characterizations. Y.Y. and C.L. prepared the cryo-EM samples. C.L. collected cryo-EM data. C.L. and Y.Y. processed the cryo-EM data and performed model building and structural analyses. C.L. and Y.Y. wrote the manuscript.

## Competing interests

The authors declare no competing interests.
