## [Transparent Peer Review file · Nature Communications]

Structural and catalytic diversity of coronavirus proofreading exoribonuclease

Corresponding Author: Dr Chang Liu

Version 0:

Reviewer comments:

Reviewer #1

(Remarks to the Author)

Dear Li, Liu and co-authors,

I have evaluated your manuscript entitled "Structural and Catalytic diversity of coronavirus proofreading exoribonuclease". The cryo-EM methodology is sound and the data has been presented meticulously. However, I do have some comments which should be addressed for a better understanding, accuracy and completeness of the data.

Comments-

1. The methods section should include a description of how the particles were picked using each method (blob picker and topaz) and what parameters including topaz models were used. How many particles were involved at each stage of the picking?
2. A representative micrograph should be shown for each of the complexes in the Extended data Figs (1-4).
3. Particles corresponding to each 3D class should be stated in all the figures (Extended data Fig 1-4). Criterion for choosing a specific 3D class for further refinement should be provided. For example, in Extended Data fig. 1., why was Class 3 chosen over Class1 (dimer) for further refinement?
4. The r. m. s deviations of the bond angles and bond length should be improved for the models specifically the MERS-Cov ExoNT20-P14-U structure.
5. What maps were involved in determining the model resolution (Extended Data Table 1)? A statement citing the map (deepenhancer maps etc.) should be reported explicitly in the legend of Extended Data Fig. 5.
6. The axes for each of FSC plot need to be labeled in all the figures (Extended Data Figs. 1-4).
7. To ensure scientifically accurate visualization, a scale bar should be added to the 2D class averages in Extended Data fig 1-4.
8. The cryo-EM maps which are deposited in the EMDB database should include the original maps in addition to the ones with density modification/ML based methods.

Reviewer #2

(Remarks to the Author)

Review Structural and catalytic diversity of coronavirus proofreading exoribonuclease

The paper by Li and collaborators presents the structure of Middle East Respiratory Syndrome Coronavirus (MERS-CoV) nsp14/nsp10 in complex with several dsRNA and a comparative structural and biochemical analyses with SARS-CoV-2. The results present a lower catalytic activity of MERS-CoV ExoN compared to that of SARS-CoV-2 ExoN, and proposed that it represents a hallmark of the divergence between ExoNs subgenera. They proposed a structural analysis to support these biochemical observation. They also defined the nucleotide recognition mechanism, the substrate preference and its specificity. The work is well executed and the results are interesting however several points needs to be clarified to be fully convincing. As presented few conclusions are overreaching.

Remarks regarding the structure of MERS-CoV and its analysis :

* Structures are informative and interesting to look at ; they contribute to a better understanding of the overall mechanism. The lacking of the second ion is indeed due to the mutation. The two ions are critical for organising the substrate and the water molecule in the cavity, therefore its lacking could also affect the final positioning of the water molecules involved in the catalytic process, relaxing the structure and generating the observed distortion described in Figure 2.

* A comparison of all the available ExoN of Sars-CoV2 (PDBs: 7QGI 9FZK, 9FW2, 9FZ4, 7N0C 7N0B) shows that a wide range of positioning of the loop harbouring H268, and that most of the structures of SARS-CoV-2 available are more similar to the open one of MERS CoV. All the relaxed loop of SARS-CoV-2 lacks the second ion (or both). Therefore the analysis suggesting that it is due to the subgenera does not hold. To reach such a conclusion a WT structure of MERS-CoV in complex with RNA where one could observed such distortion would be needed .
I suggest that the conclusion of the analysis (and discussion) to be changed accordingly.

Remarks regarding the catalytic activity of MERS-CoV and its analysis :

* Basic activity presented in Figure 1 is surprising as it takes up to 5 minutes to achieve degradation. How this substrate would do against ExoN SARS-CoV-2?

- In similar experimental conditions we need to see basal activity on a non mutated dsRNA substrate of similar length to assess if the enzyme actually has a proper activity. Have it controlled/compared to SARS CoV-2 ExoN.

* Also I would check different concentration of MG ions for optimal reaction condition. the 4mM Mg condition is great for comparison but from one to another enzyme the optimal concentration ion condition can change. in vivo this is a non issue but in vitro it can drive conclusions in the wrong direction. Data should be check and added to the result section.

Remarks regarding the Nucleotide recognition substrate preference and specificity of MERS-CoV :

This part of the study is well executed, very interesting and informative in respect of coronaviruses ExoN mode of action.

Minor remarks

Data availability section is not completed PDB/EMD CODE are under XXX

What are the noteworthy results?

Several structures of great interest of MERS-CoV and a detailed analysis of its molecular mode of action.

Will the work be of significance to the field and related fields? How does it compare to the established literature? If the work is not original, please provide relevant references.

Yes it is very important piece of work and it does cite properly the literature

Does the work support the conclusions and claims, or is additional evidence needed?

The first part of work needs more work while being well executed it lacks few controls and conclusions on the structure and subgenera specificity are overreaching. The second part of the paper structural and biochemical analysis and conclusions are well supporting the claims and results.

Are there any flaws in the data analysis, interpretation and conclusions? Do these prohibit publication or require revision?

yes (sees above) it will need to be revised

Is the methodology sound? Does the work meet the expected standards in your field?

Yes , Yes

Is there enough detail provided in the methods for the work to be reproduced?

Yes

Version 1:

Reviewer comments:

Reviewer #1

(Remarks to the Author)

The authors have addressed all my concerns and comments. I believe that the manuscript has improved overall. The structures and analyses are all well done.

Reviewer #2

(Remarks to the Author)

The authors have answered all the comments and concerns.

The article is fit for publication

Response to reviewers

We thank the reviewers for their many helpful suggestions and comments. Addressing these points has allowed us to strengthen the manuscript through the addition of new data and corrections and clarifications in the text. Below, we provide a detailed response in blue to all reviewers' comments.

Reviewer #1 (Remarks to the Author):

Dear Li, Liu and co-authors,

I have evaluated your manuscript entitled "Structural and Catalytic diversity of coronavirus proofreading exoribonuclease". The cryo-EM methodology is sound and the data has been presented meticulously. However, I do have some comments which should be addressed for a better understanding, accuracy and completeness of the data.

Response: We thank the reviewer for their appreciation of the quality of our work. As detailed below, we have addressed all the comments raised by the reviewer and improved the accuracy and completeness of the results and data presentation.

Comments-

1. The methods section should include a description of how the particles were picked using each method (blob picker and topaz) and what parameters including topaz models were used. How many particles were involved at each stage of the picking?

Response: We appreciated the reviewer's point. We have added the information regarding the procedures of particle picking using blob picker and Topaz picker in the "Cryo-EM image processing" section of Methods:

"Dose-fractioned cryo-EM movies were imported into cryoSPARC³⁹ for patch-based motion correction and patch-based CTF estimation, followed by blob picking and Topaz picking⁴⁰. Blob picking used a minimum particle diameter of 125 Å and a maximum particle diameter of 175 Å for all four cryo-EM datasets and yielded 618,359 particles for the ExoN•T20P15 complex dataset, 609,380 particles for the ExoN•T20P14-U complex dataset, 1,075,145 particles for the ExoN•T20P14-A complex dataset, and 1,050,042 particles for the ExoN•T20P14-G complex dataset, respectively. For Topaz picking, a model was generated by training for 30 epochs with a subset of ~5,000 particles from blob picking for each dataset. The generated Topaz picking model was then used for Topaz Extract job to pick particles from the entire dataset. Topaz picking yielded 687,267 particles for the ExoN•T20P15 complex dataset, 306,358 particles for the ExoN•T20P14-U complex dataset, 321,530 particles for the

ExoN•T20P14-A complex dataset, and 682,219 particles for the ExoN•T20P14-G complex dataset, respectively.”

2. A representative micrograph should be shown for each of the complexes in the Extended data Figs (1-4).

Response: We have included a representative micrograph for each dataset as panel **b** in Supplementary Figs. 1–4.

3. Particles corresponding to each 3D class should be stated in all the figures (Extended data Fig 1-4). Criterion for choosing a specific 3D class for further refinement should be provided. For example, in Extended Data fig. 1., why was Class 3 chosen over Class1 (dimer) for further refinement?

Response: We thank the reviewer for this comment. We have added the percentage for each 3D class in the cryo-EM data processing flow charts in Supplementary Fig. 1–4 (due to space restraints, we were not able to state the particle number for each 3D class in the figures). We have also added the information regarding the criterion for choosing a specific 3D class for further refinement in relevant figure legends. For example, we have added the following text in the legend of Supplementary Fig. 1d:

“Heterogeneous refinement with four ab-initio reconstructed maps in cryoSPARC generated four major 3D classes, with two of them showing structural features corresponding to a dimeric and monomeric forms of ExoN, respectively. The particles from the two 3D classes were separately subjected to another round of 3D classification. The particles from the predominant 3D class, which shows the best map features, were selected for iterative non-uniform refinement, CTF refinement, and Bayesian polishing.”

4. The r. m. s deviations of the bond angles and bond length should be improved for the models specifically the MERS-Cov ExoNT20-P14-U structure.

Response: We thank the reviewer for bringing this issue to our attention. We have fixed the bond angle and bone length outliers in the MERS-CoV ExoN•T20P14-U complex structure, re-performed the validation, and updated the statistics in Supplementary Table 1. The updated r.m.s. deviation of bond lengths and bond angles for the MERS-CoV ExoN•T20P14-U complex model is 0.004 Å and 0.942°.

5. What maps were involved in determining the model resolution (Extended Data Table 1)? A statement citing the map (deepemhancer maps etc.) should be reported explicitly in the legend of Extended Data Fig. 5.

Response: The model-map resolution for each reported structure was determined using the Resolve density-modified map, which was also used for real-space refinement. We have added such information in the “Cryo-EM model building and refinement” section of the Methods:

“All atomic models were refined against their respective Resolve density-modified maps using Phenix real-space refinement⁵³ with secondary structure restraints, rotamer restraints, and Ramachandran restraints. The final structures were validated with MolProbity⁵⁴. The Resolve density-modified map was used in validation to determine the model-map resolution for each structure.”

In addition, we have added the statement of the map used to prepare the figures in the legend of Supplementary Fig. 5:

“(a) Orientation diagnosis and histograms of 3D FSC plots for the raw cryo-EM maps. (b) Model-map FSC curves of five atomic structures and their corresponding Resolve density-modified cryo-EM maps from this study were generated from Phenix comprehensive validation results. The model-map resolution for each structure at FSC = 0.5 cutoff is indicated in the figure and summarized in Supplementary Table 1. (c) Cryo-EM densities of the DeepEMhancer-processed map superimposed on a structural model of representative regions of MERS-CoV ExoN•RNA complexes determined in this study.”

6. The axes for each of FSC plot need to be labeled in all the figures (Extended Data Figs. 1-4).

Response: We apologize. The axes for all FSC plots are now clearly labeled in the revised manuscript.

7. To ensure scientifically accurate visualization, a scale bar should be added to the 2D class averages in Extended Data fig 1-4.

Response: We thank the reviewer for raising this point. We have added a scale bar to each 2D class averages panel in Supplementary Fig. 1–4.

8. The cryo-EM maps which are deposited in the EMDB database should include the original maps in addition to the ones with density modification/ML based methods.

Response: We thank the reviewer again for this comment. We have included the original raw map, Resolve density-modified map, and DeepEMhancer-processed map for each cryo-EM map deposition. PDB and EMDB access codes for each deposition are provided in the “Data availability” section and Supplementary Table 1.

Reviewer #2 (Remarks to the Author):

1. # Review Structural and catalytic diversity of coronavirus proofreading exoribonuclease

The paper by Li and collaborators presents the structure of Middle East Respiratory Syndrome Coronavirus (MERS-CoV) nsp14/nsp10 in complex with several dsRNA and a comparative structural and biochemical analyses with SARS-CoV-2. The results present a lower catalytic activity of MERS-CoV ExoN compared to that of SARS-CoV-2 ExoN, and proposed that it represents a hallmark of the divergence between ExoNs subgenera. They proposed a structural analysis to support these biochemical observation. They also defined the nucleotide recognition mechanism, the substrate preference and its specificity. The work is well executed and the results are interesting however several points needs to be clarified to be fully convincing. As presented few conclusions are overreaching.

Response: We very much appreciate the reviewer's positive comment about the quality of our work and novelty of our findings. As detailed below, we have performed additional experiments and made changes to the text as suggested by the reviewer to address the concerns regarding few overreaching conclusions.

2. ### Remarks regarding the structure of MERS-CoV and its analysis :

* Structures are informative and interesting to look at ; they contribute to a better understanding of the overall mechanism. The lacking of the second ion is indeed due to the mutation. The two ions are critical for organising the substrate and the water molecule in the cavity, therefore its lacking could also affect the final positioning of the water molecules involved in the catalytic process, relaxing the structure and generating the observed distortion described in Figure 2.

* A comparison of all the available ExoN of Sars-CoV2 (PDBs: 7QGI 9FZK, 9FW2, 9FZ4, 7N0C 7N0B) shows that a wide range of positioning of the loop harbouring H268, and that most of the structures of SARS-CoV-2 available are more similar to the open one of MERS CoV. All the relaxed loop of SARS-CoV-2 lacks the second ion (or both). Therefore the analysis suggesting that it is due to the subgenera does not hold. To reach such a conclusion a WT structure of MERS-CoV in complex with RNA where one could observed such distortion would be needed .

I suggest that the conclusion of the analysis (and discussion) to be changed accordingly.

Response: We thank the reviewer for raising this important point. We agree that the current evidence is not sufficient to support the cross-subgenus divergence of the ExoN active site conformation and therefore have removed such claims and related discussion and figure panel in the revised manuscript.

In terms of the positioning of H268 in ExoN, the catalytic incompetent conformation of H268 in most SARS-CoV-2 ExoN structures is likely due to the absence of an RNA substrate (we note that PDBs 7QGI, 9FZK, 9FW2, and 9FZ4 were all determined without an RNA substrate). Upon the binding of an RNA substrate, H268 of SARS-CoV-2 nsp14 shifts towards the scissile phosphate and completes the ExoN active site (PDBs 7N0B, 7N0C, 7N0D, and 9Q1J). In particular, even when the active site mutation E191A was introduced into SARS-CoV-2 ExoN, the positionings of nsp14 H268 in the SARS-CoV-2 ExoN (E191A)•RNA complexes are highly similar to that observed in SARS-CoV-2 ExoN (WT)•RNA complex, despite the absence of the catalytic water molecule and weak coordination or even complete missing of divalent cation B. By contrast, nsp14 H268 of MERS-CoV ExoN adopts an open and catalytically inactive conformation in all five structures reported in our current study (as shown in new Supplementary Fig. 6d in the revised manuscript), regardless of the identity of the 3'-end nucleotide in the RNA substrate. Although these cryo-EM structures were determined using MERS-CoV ExoN E191A mutant, the direction comparison of these structures with those of SARS-CoV-2 ExoN•RNA complexes carrying the same ExoN active site mutation still strongly suggests that H268 of MERS-CoV ExoN is less prone than SARS-CoV-2 ExoN to adopt the optimal conformation required to perform catalysis and provides structural insights into the lower catalytic activity of ExoN from MERS-CoV than that from SARS-CoV-2.

We have changed the text accordingly in the revised manuscript (Line 131–140):

“Whereas RNA binding to SARS-CoV-2 ExoN, either the WT or E191A mutant form, induces a substantial structural rearrangement of its H268-harboring α 4- α 5 loop, facilitating the full assembly of the ExoN active site²³, the conformation of the α 4- α 5 loop in the MERS-CoV ExoN•RNA complex resembles that in the RNA-free apo SARS-CoV ExoN^{15,25} (Fig. 2c), which represents an inactivated state prior to the binding of RNA substrate²³. In addition, we modeled the position of the catalytic water molecule in the MERS-CoV ExoN•RNA complex structure based on its superimposition with the SARS-CoV-2 WT ExoN•RNA complex structure²³ (Fig. 2d). Our structural modeling suggests that H268 of MERS-CoV ExoN would be excessively distant from the catalytic water ($> 4\text{\AA}$) (Fig. 2d), hampering an efficient deprotonation of the catalytic water molecule for subsequent nucleophilic attack of the scissile phosphate²³.”

3. ### Remarks regarding the catalytic activity of MERS-CoV and its analysis :

* Basic activity presented in Figure 1 is surprising as it takes up to 5 minutes to achieve degradation. How this substrate would do against ExoN SARS-CoV-2?

Response: We thank the reviewer for the comment. In the Fig. 1c of the original manuscript, there was already significant digestion of the T20P15 substrate RNA at time points as early as 20s and 40s. However, due to the insufficient resolution of the denaturing PAGE result presented in our original figure, the substrate band was not well separated from the products

whose size is only one or two nucleotides shorter than the substrate. In the revised manuscript, we have re-performed the assay, adding more time points and SARS-CoV-2 ExoN as a control, and improved the separation resolution of the PAGE. Our result shows that MERS-CoV ExoN WT is highly active on the RNA substrate (significant cleavage products are observed starting from 20–30s), although its activity is much lower than that of SARS-CoV-2 ExoN. This observation is consistent with the results obtained in exonuclease assays performed on other RNA substrates (such as those presented in Fig. 4a).

We have included the new result as Fig. 1c and added the following text in the revised manuscript (Line 142–146):

“To assess the catalytic difference between these two coronavirus ExoNs, we first compared the digestion of MERS-CoV ExoN and SARS-CoV-2 ExoN on the T20P15 RNA. Our result shows that while MERS-CoV ExoN is highly active in cleaving the RNA substrate, its digestion rate is markedly lower than that of SARS-CoV-2 ExoN (Fig. 1c).”

4. - In similar experimental conditions we need to see basal activity on a non-mutated dsRNA substrate of similar length to assess if the enzyme actually has a proper activity. Have it controlled/compared to SARS-CoV-2 ExoN.

Response: We agree with the reviewer and such experiments were indeed included in our manuscript (Fig. 4a,b and Supplementary Fig. 7a,b), where we examined and compared the digestion of an 5'-FAM-labeled non-mutated dsRNA substrate bearing a 3'-end U:A, A:U, G:C, or C:G base pair (in addition to dsRNA with a 3'-end U:C mismatch) by MERS-CoV and SARS-CoV-2 ExoN, respectively. Our results showed that MERS-CoV ExoN is active on all these non-mutated dsRNA substrates but exhibits a lower digestion rate than SARS-CoV-2 ExoN does.

5. * Also I would check different concentration of Mg ions for optimal reaction condition. The 4mM Mg condition is great for comparison but from one to another enzyme the optimal concentration ion condition can change. In vivo this is a non-issue but in vitro it can drive conclusions in the wrong direction. Data should be checked and added to the result section.

Response: We appreciate the reviewer's insightful and constructive suggestion for examining the Mg²⁺ concentration that is optimal for the activities of ExoNs from MERS-CoV and SARS-CoV-2. We have measured the RNA substrate digestion by MERS-CoV ExoN or SARS-CoV-2 ExoN in the presence of Mg²⁺ concentrations ranging from 1 mM to 16 mM and found that the activity of SARS-CoV-2 ExoN is much higher than that of MERS-CoV ExoN at each of the tested Mg²⁺ concentrations.

We have included the new result as Fig. 2f and added the following text in the revised manuscript (Line 154–162):

“Furthermore, to examine the effect of Mg^{2+} concentration on the catalytic activities of MERS-CoV and SARS-CoV-2 ExoNs, we determined the digestion rate constants (k) of the two coronavirus ExoNs across a range of Mg^{2+} concentrations. While both ExoNs experience a gradual boost of their catalytic activity as Mg^{2+} concentration increases from 1 mM to 16 mM, the digestion rate constant of SARS-CoV-2 ExoN is substantially higher than that of MERS-CoV ExoN at each of the tested Mg^{2+} concentrations (Fig. 2f). Therefore, our result indicates that the considerably lower catalytic activity of MERS-CoV ExoN than that of SARS-CoV-2 ExoN is not because of its preference to a different Mg^{2+} concentration, but rather likely stems from the suboptimal conformation of its catalytic residue H268 (Fig. 2b).”

6. ### Remarks regarding the Nucleotide recognition substrate preference and specificity of MERS-CoV :

This part of the study is well executed, very interesting and informative in respect of coronaviruses ExoN mode of action.

Response: We are very grateful to the reviewer for the positive comment on the quality of our work and significance of our findings.

7. ### Minor remarks

Data availability section is not completed PDB/EMD CODE are under XXX

Response: We apologize. We have deposited the structural models and cryo-EM maps in the Protein Data Bank and Electron Microscopy Data Bank, respectively, for all five cryo-EM structures reported in this study and included the accession codes in the “Data availability” section of the revised manuscript:

“Atomic coordinates of five structures determined in this study have been deposited in the Protein Data Bank with accession codes 9YCK (MERS-CoV ExoN•T20P15 complex, monomeric form), 9YCL (MERS-CoV ExoN•T20P15 complex, dimeric form), 9YCM (MERS-CoV ExoN•T20P14-U complex), 9YCN (MERS-CoV ExoN•T20P14-A complex), and 9YCO (MERS-CoV ExoN•T20P14-G complex). The cryo-EM maps have been deposited in the Electron Microscopy Data Bank with accession number EMD-72775 (MERS-CoV ExoN•T20P15 complex, monomeric form), EMD-72776 (MERS-CoV ExoN•T20P15 complex, dimeric form), EMD-72777 (MERS-CoV ExoN•T20P14-U complex), EMD-72778 (MERS-CoV ExoN•T20P14-A complex), and EMD-72779 (MERS-CoV ExoN•T20P14-G complex).”

—

8. ### What are the noteworthy results?

Several structures of great interest of MERS-CoV and a detailed analysis of its molecular mode of action.

Will the work be of significance to the field and related fields? How does it compare to the established literature? If the work is not original, please provide relevant references.

Yes it is very important piece of work and it does cite properly the literature

Does the work support the conclusions and claims, or is additional evidence needed?

The first part of work needs more work while being well executed it lacks few controls and conclusions on the structure and subgenera specificity are overreaching. The second part of the paper structural and biochemical analysis and conclusions are well supporting the claims and results.

Are there any flaws in the data analysis, interpretation and conclusions? Do these prohibit publication or require revision?

yes (sees above) it will need to be revised

Is the methodology sound? Does the work meet the expected standards in your field?

Yes , Yes

Is there enough detail provided in the methods for the work to be reproduced?

Yes

Response: We very much appreciate the reviewer's enthusiastic comments about the significance of our study, the soundness of our methodology, and the quality of our results. As described above, we have performed additional experiments and revised the text in the manuscript to address the reviewer's concerns regarding the lack of few controls and few overreaching conclusions. We believe that the revision has greatly strengthened the manuscript. We thank the reviewer again for their constructive comments.